# Fucoidan from *Undaria pinnatifida* Ameliorates Epidermal Barrier Disruption via Keratinocyte Differentiation and CaSR Level Regulation

**DOI:** 10.3390/md17120660

**Published:** 2019-11-24

**Authors:** Yu Chen, Xuenan Li, Xiaoshuang Gan, Junmei Qi, Biao Che, Meiling Tai, Shuang Gao, Wengang Zhao, Nuo Xu, Zhenlin Hu

**Affiliations:** 1College of Life and Environmental Sciences, Wenzhou University, Wenzhou 325035, China; chenyu5144@hotmail.com (Y.C.); qjm1357@163.com (J.Q.); 2Institute of Life Sciences, Wenzhou University, Wenzhou 325035, China; lixuenan1630@163.com (X.L.); gaoshuangphu@163.com (S.G.); zwg123@wzu.edu.cn (W.Z.); 3Infinitus (China) Company Ltd, Guangzhou 510000, China; Sunny.Gan@infinitus-int.com (X.G.); Bill.Che@infinitus-int.com (B.C.); Sunny.Tai@infinitus-int.com (M.T.)

**Keywords:** epidermal barrier, UPF, CaSR, ERK, p38

## Abstract

The epidermal barrier acts as a line of defense against external agents as well as helps to maintain body homeostasis. The calcium concentration gradient across the epidermal barrier is closely related to the proliferation and differentiation of keratinocytes (KCs), and the regulation of these two processes is the key to the repair of epidermal barrier disruption. In the present study, we found that fucoidan from *Undaria pinnatifida* (UPF) could promote the repair of epidermal barrier disruption in mice. The mechanistic study demonstrated that UPF could promote HaCaT cell differentiation under low calcium condition by up-regulating the expression of calcium-sensing receptor (CaSR), which could then lead to the activation of the Catenin/PLCγ1 pathway. Further, UPF could increase the expression of CaSR through activate the ERK and p38 pathway. These findings reveal the molecular mechanism of UPF in the repair of the epidermal barrier and provide a basis for the development of UPF into an agent for the repair of epidermal barrier repair.

## 1. Introduction

The outermost layer or epidermis of the skin serves to perform the majority of the functions of the skin such as the first line of defense against the environment. This epidermis not only functions in defense against toxic and pathogenic agent entry but also suppresses the loss of water, ions, and metabolites. Environmental changes, physical damage, and glucocorticoid drug abuse affect the epidermal barrier functions while disruption of the epidermal barrier can lead to the incidence of a variety of skin diseases such as psoriasis, atopic dermatitis, and infections by bacteria and viruses [1,2]. There are several layers in the epidermis with the layers from inside-out being basal layer, spinous layer, granular layer and the stratum corneum (SC). The healthy epidermis is a self-renewing tissue that renews itself entirely in around 28 days on the basis of balanced division and differentiation to maintain the barrier properties [1]. Approximately 90%–95% of epidermal cells are keratinocytes making up the chief type of cell in the epidermal barrier. These cells undergo division in the basal layers to later differentiate across the epidermis to become SC cells that are enucleated and flattened which will desquamate eventually. The process of differentiation in the epidermis involves the development of numerous layers of keratinocytes from the stratum basale to the stratum spinosum and the stratum corneum, expressing distinctive marker genes at each stage of differentiation [3,4]. The keratins K5 and K14 are expressed at the level of in the stratum basale that later are transformed to the keratins K1, K10 and involucrin in the stratum spinosum, while the chief elements of the cornified envelope (CE), loricrin and filaggrin, are also expressed in the stratum granulosum [5]. The granular to cornified transition sees CE formation as the principal change in structure [6,7]. Interference with the epidermal barrier integrity or function may disturb the balance between proliferation and differentiation, leading to excessive proliferation as well as abnormal differentiation, subsequently leading to epidermal hyperplasia [8].

Early and current treatment approaches to wounds or skin issues have resorted to the use of multifunctional compounds called polysaccharides that as active secondary products have piqued great attention. Keratinocytes proliferation and differentiation has been regulated by some polysaccharides such as those from kiwi fruits or Okra fruit that could stimulate the proliferation of human keratinocytes [9]. Milk oligosaccharides could induce keratinocytes differentiation through augmented expression of involucrin. Polysaccharides form *Typha latifolia* fruits have a strong stimulatory activity on keratinocytes proliferation as well as differentiation [10,11]. Whereas the mechanism of these polysaccharides promoting keratinocytes proliferation and differentiation is unclear. Brown marine algae are the source of fucoidans that are described as sulfated polyfucose polysaccharides. Many properties of fucoidan have been shown by research including targeting coagulation [5], anti-tumor, immunomodulatory, antioxidant, and anti-inflammatory effects [12,13,14]. It was also reported that fucoidan could contribute to the reconstruction of skin while boosting type I procollagen production and inhibiting matrix metalloproteinase MMP-1 levels induced by UV-B [15,16]. Additionally, dermal wound healing has been reported by this polysaccharide [17].

Therefore, this work was aimed at investigating the effect of fucoidan from *Undaria pinnatifida* (UPF) on barrier repair of the damage induced by tape-stripping along with the exploration of the putative underlying mechanisms of repair.

## 2. Results

### 2.1. The Molecular Weight and Monosaccharide Composition

The GPC-MALLS determination showed the molecular weight of UPF to be 171KD (Figure 1A). The distribution of the monosaccharide molar ratio of UPF is presented as mannose: rhamnose: galactose: fucose = 11.7:4.14:12.7:7.49 (Figure 1B).

### 2.2. UPF Could Promote the Epidermal Barrier Recovery

To investigate the influence of UPF on the recovery of epidermal barrier disruption, tape stripping, that can damage the stratum corneum extensively, was employed to induce the acute barrier damage in mice. TEWL is the passage of water into the atmosphere from the stratum corneum under normal conditions. Increased TEWL was associated with increased skin permeability and chemical absorption and hence, damage, thus, measurement of the TEWL is a marker for skin barrier function. Scab formation, exfoliation, and hair growth occurred earlier in the injured area of back skin in UPF treated mice as against controls (Figure 2A). The TEWL results showed that the recovery rate of barrier disruption in mice exposed to UPF (0.5%, 5%) was evidently more rapid than that in mice treated without UPF, especially at 48 h and 72 h, the barrier recovery rate of the 0.5% and 5% groups were significantly accelerated compared with vehicle control before 72 h, only 5% UPF treated mice showed significant improvement in repair rate at 72 h and 84 h (Figure 2B). H&E staining was utilized to observe hyperplasia in the epidermis due to the tape stripping. The results revealed that the epidermal thickness of 0.5% and 5% UPF treated mice was much less than that of model mice, indicating that the UPF could alleviate the symptom of epidermal hyperplasia during the recovery process of the barrier disruption (Figure 2C). The immunohistochemical was used to detect the expression of differentiation markers such as involucrin and filaggrin. These results demonstrated that the expression of involucrin and filaggrin in the dorsal skin of 0.5% and 5% UPF-treated mice were significantly improved compared with the vehicle group, revealing that the UPF could promote the epidermal differentiation during recovery process to alleviate the epidermal hyperplasia (Figure 2D). 

### 2.3. Keratinocytes Differentiation and Sustained Ca^2+^ Concentration by UPF in HaCaT Cells

As discussed earlier, the chief cells in the skin barrier are make up the epidermal structure and the cornified cell layers exposed to the surroundings directly. The HaCaT possesses complete epidermal differentiation ability and similar biological characteristics with that of primary keratinocytes to, hence, serve as a fitting system to study epidermal differentiation in vitro. The qPCR results demonstrated that UPF (10, 20, 50 μg/mL) could dose-dependently increase the mRNA expressions of differentiation markers, including involucrin and filaggrin in HaCaT cells at 72 h (Figure 3A). UPF could also promote the protein expression of involucrin and filaggrin in HaCaT cells (Figure 3B). As shown in Figure 3C,D, UPF could augment both the mRNA levels and protein expression of involucrin and filaggrin in NHEK cells at 72 h, those results are completely consistent with that in HaCaT cells, indicating that the HaCaT cells can be used to study epidermal differentiation in vitro in this research. The BrdU assay revealed the promotion of HaCaT cell proliferation by UPF in proportion to the dose of the latter at 24 h, but with no effect at 48 h and 72 h at the tested concentrations (Figure 3E).

The process of keratinocytes commitment to differentiation across all the epidermal layers involves a vital role of calcium. It is well established that extracellular calcium (Ca_o_) concentrations above 10 mM induces differentiation in HaCaT cells, whereas the present study demonstrated that the UPF could promote differentiation of HaCaT cell under normal culture conditions (calcium concentration is 1.8 mM). A primary role of Ca^2+^ to regulate the differentiation and proliferation of keratinocytes is known with a higher release of intracellular calcium (Ca_i_) from storage sites in response to elevated Ca_o_ levels, hence, increasing the expression of genes involved in keratinocytes differentiation [18]. This Ca_o_-induced response of keratinocytes involves several steps of transient sustained with the latter vitally functioning in the case of differentiation [19]. To determine whether UPF can affect the sustained Ca_i_ concentration in HaCaT cells, cells were exposed to UPF for 24 h under normal culture conditions, and then the Ca_i_ concentration was measured by confocal microscopy. As shown in Figure 3F, treatment of the cells with UPF evidently augmented the Ca_i_ concentration against the control, suggesting that UPF may promote differentiation via augmenting the sensitivity of keratinocytes to calcium. Further studies are required to unveil the details of this process.

### 2.4. UPF Could Boost the Expression of CaSR and Activate CaSR Mediated Signaling Pathway

Alterations in Ca_i_ levels are sensed by a family of G protein-coupled receptors called calcium-sensing receptors (CaSR) [20]. To investigate the effects of UPF on mRNA and protein expression of CaSR, HaCaT cells were incubated with UPF at 10, 20 and 50 µg/mL for 12 h, 24 h or 48 h, followed by qPCR and Western blot analysis, respectively. The result demonstrated that UPF could dose dependently elevate the mRNA expression of CaSR at 24 h and 48 h (Figure 4A). The Western blot result showed that UPF could increase the protein expression of CaSR at 12 h, 24 h or 48 h (Figure 4B) in a dose-dependent manner. As shown in Figure 4C, when activated by calcium, the CaSR, in turn activates Gαq (a G protein), that results in the activation of PLC-β (phospholipase C-β). This activates PLC-β and converts phosphatidyl inositol-4,5-bisphosphate (PIP_2_) into inositol 1,4,5-trisphosphate (IP_3_) and diacylglycerol (DAG). IP_3_ mediates the release of calcium from endoplasmic calcium stores such as ER and Golgi by activating dependent on the ligand, resulting in an acute increase in Ca_o_ [19,21]. This is the mechanism associated with the rise in Ca^2+^ levels described. RhoA is activated by the stimulation of CASR by G_12/13_, filamin A, and RhoGEF. This activation of PI3K is due to tyrosine phosphorylation by Fyn and Src, Src family kinases of the catenins at the cell membrane to form E-cadherin/catenin complex [22], which in turn activates PLC-γ1, and converts PIP_2_ into IP_3_, which results sustained increasing intracellular Ca^2+^ concentration [23,24]. The sustained Ca_i_ as discussed earlier is vitally involved in subsequent ketatinocyte differentiation [25,26]. When Ca_i_ is increased, the expression of Wnt5a is elevated and then secreted to the outside of keratinocytes, binds to its receptor Fz-LRP5/6 subsequently, further to stimulate β-catenin entry into the nucleus, promoting the expression of differentiation markers eventually [27]. Treatment of HaCaT cells with UPF significantly induced the phosphorylation of PLC-γ1 and p120-catenin at 12 h, in addition, the protein expression levels of β-catenin in the nucleus was also increased, suggesting that UPF could activate CaSR-mediated signaling pathway (Figure 4D). HaCaT cells were exposed to UPF with or without NPS-2143 (CaSR antagonists) for 72 h, the Western blot results showed that NPS-2143 blocked CaSR-mediated signaling pathway and inhibit the involucrin and filaggrin expression induced by UPF (Figure 4E). As shown in Figure 4F, the qPCR results also confirmed that UPF could not elevate the mRNA level of involucrin and filaggrin in HaCaT cells once the CaSR pathway was blocked indicating that UPF may induce the HaCaT differentiation through CaSR.

### 2.5. UPF Activated ERK and p38 Signaling Pathway

The activation of ERK and p38 signaling pathway could trigger the keratinocytes differentiation as well as the expression of markers mediated by Ca^2+^ [28,29,30,31]. The increased amounts of CaSR are allowed for by UPF via ERK and p38 signaling pathways. UPF exposure of HaCaT cells caused a significant phosphorylation of ERK and p38 at 12 h (Figure 5A), compared with the control, suggesting that UPF may modulate cell differentiation via ERK and p38 signaling pathway activation. Blocking of this pathway also targets the UPF induced phosphorylation of PLCγ1 and p120-catenin, the relative expression levels of phosphorylation of PLCγ1 and p120-catenin were not elevated, indicating that UPF might trigger the activation of CaSR-mediated PLCγ1/p120-catenin signaling pathway via ERK and p38 pathway (Figure 5B,C). Figure 5D showed the increase in CaSR expression levels by UPF was markedly attenuated by blocking the activation of ERK and p38. The mRNA levels of involucrin and filaggrin in HaCaT cells were also attenuated once the ERK and p38 signaling pathway was blocked (Figure 5E), suggesting that UPF stimulates CaSR expression via activating the ERK and p38 pathways, thereby promoting the differentiation of keratinocytes.

## 3. Discussion

The maintenance of the epidermal barrier involves a balanced keratinocytes proliferation and differentiation; these cells serve as the major cell type of the epidermis making up its structure as well as the barrier [3]. In the epidermis, division of the keratinocytes is limited to the basal cell layers. During differentiation, keratinocytes move upwards to form stratum spinosum, stratum granulosum and finally, in the stratum corneum, become the terminally differentiated corneocytes.

Once a functional epidermal barrier is formed, the stratum corneum is desquamated and replenished through keratinocytes terminal differentiation. The imbalance of proliferation and differentiation processes may lead to some skin diseases. The decreased rate of stratum corneum desquamation leads to ichthyoses whereas a higher proliferation rate is seen in psoriasis [32,33,34]. This work involved the construction a mouse model of epidermal barrier disruption by tape-stripping. We first found that UPF extracted from *Undaria pinnatifida* could accelerate epidermal repair compared with the vehicle group. The disruption of epidermal barrier resulted in the epidermal hyperplasia which could be alleviated by UPF in our study. Given that a disruption in epidermal barrier is linked to higher TEWL, this was estimated to reflect the loss in water and epidermal changes. A point of interest uncovered in this work was lowered TEWL by UPF in proportion to the dose, indicating that the UPF could promote the recovery of epidermal barrier disruption of mouse. The spontaneously immortalized human HaCaT cell line is derived from normal adult skin and has been widely used in the study of keratinocytes and epidermal biology because of its intact epidermal differentiation ability and similar biological characteristics to primary keratinocytes [35,36]. Additionally, HaCaT is stable and easy to culture, so we select HaCaT as cell model in this research to study the effect of UPF on epidermal differentiation in vitro. Further mechanism study in vitro demonstrated that the UPF could promote keratinocytes differentiation.

Studies have shown the regulatory involvement of calcium in keratinocytes differentiation and initiating proliferation. A calcium gradient is observed in the epidermis with the levels varying from low amounts in basal and spinous layers that slowly increases in the stratum granulosum, to dip in the stratum corneum [37]. The low amounts in the basal layer allows for the proliferation of keratinocytes with higher levels influencing the various markers of the cells [38]. This calcium gradient thus functions in the permeability barrier of the epidermis. Lowered transepidermal water loss at a low rate at a sustained rate allows for this calcium gradient [39].

The use of topical solvent or tape-stripping causes an increased water movement, followed by the flow of calcium through SC to cause the loss of extracellular calcium loss from epidermis to alter the calcium concentration in the outer epidermis from 460 ± 57 to 128 ± 14 mg/kg [40]. The loss of calcium gradient inhibits the differentiation of keratinocytes, and trigger overproliferation to compensate for the barrier disruption, resulting in epidermal hyperplasia eventually [6,18,41]. A higher calcium in the uppermost epidermis alters the body secretion in the lamellae to delay the repair of the skin barrier [42]. Therefore, methods to stimulate the differentiation of keratinocytes may be important strategies during the epidermal barrier recovery process after disruption.

While Ca_o_ concentrations above 10 mM induce HaCaT cell differentiation, the present study demonstrated that the UPF could promote both the proliferation and differentiation of keratinocytes under 1.8 mM calcium concentration, indicating of this promotion at low calcium concentration. This may be a vital mechanism of UPF in promoting the recovery of epidermal barrier disruption. Elevating Ca_o_ causes the stimulation of a sustained increase in Ca_i_ to promote important signaling events to in turn promote the differentiation of keratinocytes. This higher Ca_i_ concentration is attributed to the release of Ca^2+^ from internal stores and plasma membrane channels [43]. The confocal microscopy result in this report revealed that UPF could elevate sustained Ca_i_ in proportion to the concentration suggestive of promoted differentiation by UPF via boosted calcium sensitivity.

The involvement of protein G-coupled receptors called CASR functions in Ca_o_ for differentiation of keratinocytes, as its deletion attenuated the response to Ca_o_ and decreased the keratinocytes differentiation [44]. In addition, the keratinocyte-specific CaSR knockout mice showed that an absence of the calcium gradient in the epidermis with higher division and lowered the expression of the differentiation markers, suggesting the vital role of CaSR in epidermal differentiation and barrier function [37]. This work evaluated the ability of UPF to increase the sensitivity of the keratinocytes to calcium in relation to CaSR. Our data showed that UPF could elevate the level of CaSR at 12 h in a dose-dependent manner. The UPF could induce the phosphorylation of p120-catenin and PLCγ1, indicating that UPF could activate the CaSR mediated signaling pathway. Thus, UPF may increase the sensitivity of keratinocytes to calcium through an increase of CaSR expression, further to induce the differentiation under low calcium levels, to promote its effect eventually on epidermal barrier disruption. Inactivation of MAPK signaling pathway can decrease the expression of caspase-14, and further aggravate the impaired skin barriers [45]. The differentiation of keratinocytes also involves regulation by the canonical ERK cascade of the mitogen activated protein kinase (MAPK) family [29,30]. The expression of involucrin is activated by p38 in cultured keratinocytes [31]. The expression of markers was inhibited in the case of dominant negative mutants of ERK and p38 in Ca^2+^ shifted cells showing the involvement of this pathway in marker expression mediated by Ca^2+^ [28]. UPF administration to HaCaT cells induced ERK and p38 phosphorylation at 12 h in proportion to the dose. Pretreatment of HaCaT with ERK inhibitor, and p38 inhibitor, evidently suppressed the UPF induced HaCaT differentiation, suggesting that UPF may modulate keratinocytes differentiation via this signaling pathway. Collectively, these results illustrated that the UPF could increase the expression of CaSR via the ERK and p38 pathway to increase the sensitivity of keratinocytes to calcium, attenuate the epidermal hyperplasia, and promote the recovery of epidermal barrier disruption *in vivo* eventually. The exact mechanisms necessitate the need of further research.

## 4. Materials and Methods

### 4.1. Reagents

NPS-2143, LY3214996 and SB203580 were purchased from Selleck (Shanghai, China).

### 4.2. Mice

Shanghai SLAC Laboratory Animal Co., Ltd. (Shanghai, China) was the source of ICR mice (male, 18–22 g). The Animal Care and Use Committee of Wenzhou Medical University issued approval of the use and care of animals in this work that was in lieu of the Guide for the Care and Use of Laboratory Animals from the National Institutes of Health. The maintenance of the animals was done as a 12 h light/dark cycle along with standard conditions of temperature and humidity of 22 ± 1 °C and 50 ± 10%, respectively. Acclimatization was done at least a week before studies while the nutrition was a sterilized pellet-based diet.

### 4.3. Determination of Molecular Weight

GPC-MALLS (Wyatt Technology Corporation, Santa barbara, CA, USA) was employed for evaluating the distributions of molecular weight and mass of UPF. This system employs a Waters 2690D separations module, a Waters 2414 refractive index detector (RI) and a Wyatt DAWN EOS MALLS detector. Ten milligrams of UPF was dissolved in 1 mL of 0.1 M NaNO_3_ solution that was rolled overnight so that the entire amount was dissolved. Post a four-fold dilution, a 0.45 μm nylon filter (Millipore Corp., Billerica, MA, USA) was utilized to filter the solution of which 100μL was injected. A series of OHpak SB-805 HQ, OHpak SB-806 HQ, and OHpak SB-803 HQ columns were used. The collection and analysis of data was done using Astra 4.90.08 software (Wyatt Technology Corporation, Santa barbara, CA, USA).

### 4.4. Analysis of Monosaccharides

The procedure employed for the derivatization of monosaccharide was carried out following the approach of Xia Y et al. [46]. Concisely, this involved the addition of 2 mL 4 M TFA to 10 mg of the polysaccharide sample that was kept for 8 h at 100 °C oil bath. The residual TFA was evaporated using methanol followed by freeze drying to prepare the sample for derivatization. The final concentration of PMP was 0.5 M in methanol. Ammonia (5 mL) was used for dissolving the standard sugars. The polysaccharide or standard monosaccharides were dissolved in 5 mL ammonia. Then, the 100 μL solution was mixed with 0.5 M methanol solution (100 μL) of PMP. The mixture cooled to ambient temperature after reaction for 30 min at 70 °C, then neutralized with 200 µL of formic acid. Prior to UPLC/Q-TOF-MS analysis, filtration of the solution was done using a 0.22 µm membrane followed by dilution using deionized water.

### 4.5. Epidermal Barrier Disruption

Tape stripping was done for targeting the permeability barrier of the epidermis of the mice until the transepidermal water loss (TEWL) reached 40 mg/cm^2^/hour (as measured by Vapometer Delfin Technologies, Finland). Random division of the animals was done (*n* = 3) as follows: control group (normal mice without any treatment), model group (tape stripping only), 0.5% UPF group (tape stripping + 0.5% UPF hydrogel), and 5% UPF group (tape stripping + 5% UPF hydrogel). The application of UPF in the test groups was topical on the dorsal skin. The measurement of TEWL was done at the time points 0, 6, 12, 24, 48, 72, and 84 hours at the sites of treatment sites post barrier disruption. The recovery of the barrier has been presented as the percent recovery using the expression: (TEWL immediately post disruption of the barrier – TEWL at the indicated time point)/ (TEWL immediately post disruption of the barrier –baseline TEWL) × 100%.

### 4.6. Histological Examination

To evaluate the expressions of differentiation markers in the epidermis, the mice dorsal skin were obtained at the 84 h, and was subjected to fixation using 10% neutral buffered formalin. This was followed by embedding the samples in paraffin and sliced into 6 μm thick sections. For immunohistochemistry, the sections were incubated in 3% H_2_O_2_ for 25 min and then blocked by 3% BSA for 4 h. After that, the sections were incubated overnight at 4 °C with the immunohistochemical markers: anti-Filaggrin (1:500; Absin, Shanghai, China), anti-Involucrin (1: 200; Proteintech, Wuhan, China). After washed with PBS three times, the sections were incubated with horseradish peroxidase-conjugated secondary antibodies for 4 h at 37 °C and then reacted with 3,3-diaminobenzidine (DAB). All stained sections were examined by DM3000 microscopy (Leica, Wetzlar, Germany) to assess the histological changes.

### 4.7. Cell Culture

Zhong Qiao Xin Zhou Biotechnology Co., Ltd. (Shanghai, China) was the source of HaCaT cells (cat. no., ZQ0044) and NHEK cells (cat. no., ZQ2110). HaCaT, a human keratinocyte cell line immortalized spontaneously. Dulbecco’s modified Eagle’s medium (DMEM), 10% fetal bovine serum (FBS) (both from GIBCO, Life Technologies Corporation, NY, USA), 1% each of penicillin and streptomycin were utilized for culture HaCaT at 37°C in 5% CO_2_. NHEK cells were cultured in keratinocyte serum-free medium (KSFM) containing 1% antibiotic/antimycotic (AB/AM) solution at 37 °C in a humidified atmosphere with 5% CO_2_.

### 4.8. BrdU Assay

Post the use of 96-well plates for seeding HaCaT cells, UPF was added for 24, 48, 72 h followed by the use of a BrdU assay kit (Roche, Mannheim, Germany) in accordance with the protocols of the manufacturer. In a nutshell, this involved exposing the cells for 6 h to 10 µM of BrdU at 37 °C followed by fixation using denaturing solution. Addition of anti-BrdU antibody was done for 90 min at 25 °C after which the substrate was added for 20 min and measuring the absorbance of the plate in a microplate reader at 370 nm.

### 4.9. RNA Isolation and Quantitative Real-time PCR (qRT-PCR) Analysis

Post 24 hours of culture of 1 × 10^5^ HaCaT cells or NHEK cells/well, 10, 20, 50 μg/mL of UPF was added to the cells. This was followed by extraction of total RNA utilizing the RNA isolation kit (Biomiga, San Diego, CA, USA) in accordance with the prescribed protocols of the manufacturer. PrimeScript RT reagent Kit (Takara, Dalian, China) was utilized for reverse transcription of 1 μg of total RNA/sample. An LC96 system (Roche, Basel, Switzerland) was utilized for qRT-PCR employing the SYBR Green Master Mix (Applied Biosystems, Foster City, CA, USA). The primer details are given below: Filaggrin-forward, 5′-CCCAGGTCCCATCAAGAAGA-3′ and reverse 5′-TGAGTCTGTGGAGCTGTCTG-3′; Involucrin-forward 5′-TAGAGGAGCAGGAGGGACAA-3′ and reverse 5′-GTGCTTTTCCTGCTGTTCCA-3′; CaSR-forward 5′-GGAGCAGGTGACCTTTGATGAG-3′ and reverse 5′-GAGAGGTGCCAGTTGATGATGG-3′; GAPDH-forward 5′-CAGGAGGCATTGCTGATGAT-3′ and reverse, 5′-GAAGGCTGGGGCTCATTT-3′. The comparative Ct approach was utilized to assay the mRNA relative levels while the endogenous control was GAPDH while the 2^−ΔΔCt^ method was employed to quantify the relative target gene expression.

### 4.10. Measurement of Sustained Intracellular Calcium

An eight-well ibidi plate (Dojindo, Kumamoto, Japan) was utilized to seed HaCaT cells that were exposed to UPF as mentioned above while controls received no treatment. After that HaCaT cells were loaded with 5 μM Fura-4 AM in HBSS (Life Technologies Corporation, NY, USA) for at 37 °C for 20 min. Cells were then washed three times and incubated with HBSS at 37 °C for 10 min. Fluorescence was recorded with a Ti2-E&CSU-W1 confocal microscope (Nikon, Tokyo, Japan) based on the fluorescence ratios at 494 nm (excitation) and 516 nm (emission) wavelengths, respectively.

### 4.11. Western Blotting

A six-well plate was used to culture HaCaT cells or NHEK cells for 24 h, followed by 12 h, 24 h, 48 h, or 72 h of UPF treatment at the above-mentioned concentrations along with controls. Following cell harvest, the total protein extracted from each culture was resolved on 8% SDS-PAGE. This was followed by transfer to polyvinylidene difluoride (PVDF) membrane and overnight incubation with the following antibodies at 4 °C: with rabbit anti-phospho-p120-catenin, anti-p120-catenin, anti-phospho-PLCγ1, anti-PLCγ1, anti-phospho-ERK1/2, anti-ERK1/2, anti-phospho-p38, anti-p38, anti-CaSR, anti-Involucrin or anti-tubulin antibody (Cell Signaling Technology, Beverly, MA, USA), and with anti-Filaggrin (Santa Cruz Biotechnology, Dallas, TX, USA) for overnight. Post addition of goat anti-rabbit antibody or goat anti-mouse antibody (Cell Signaling Technology, Beverly, MA, USA) respectively, the detection was done using a chemiluminescence substrate (Pierce, Rockford, IL, USA) followed by the use of Amersham Imager (GE Healthcare Biosciences, Pittsburgh, PA, USA) to record images.

### 4.12. Statistical Analysis

Means ± SEMs was used to depict all the data. One-way ANOVA was utilized to evaluate significant differences among the test samples. *p* < 0.01 and *p* < 0.05 was used for significance.

## 5. Conclusions

In summary, we demonstrated here that UPF could attenuate epidermal hyperplasia and promote the recovery of epidermal barrier disruption *in vivo*, with the involvement of the ERK and p38 pathway and augment the sensitivity of keratinocytes to calcium by increased expression of CaSR. These findings are suggestive of the potential of UPF in application as a skin repair agent.

## Figures and Tables

**Figure 1 marinedrugs-17-00660-f001:**
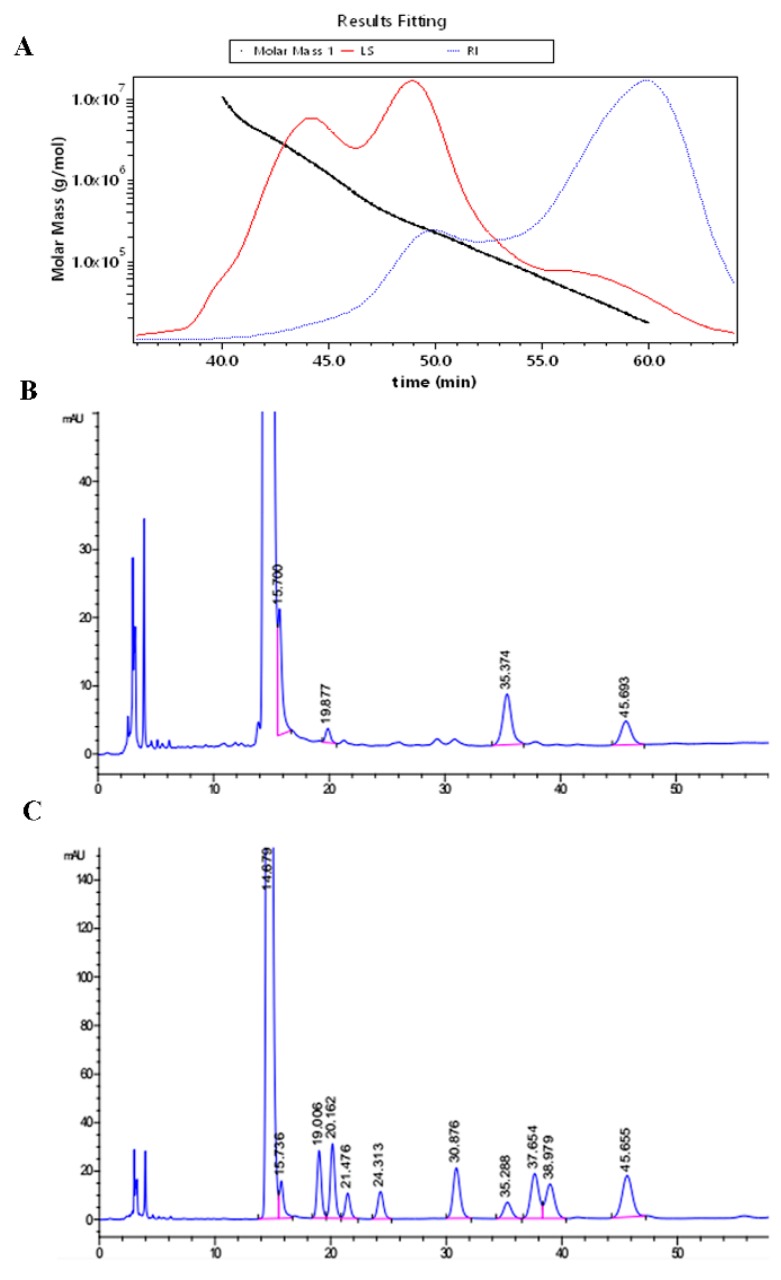
Molecular weight and monosaccharide composition of UPF. (**A**) The molecular weight and molecular mass distributions of UPF were determined by GPC-MALLS consisting of a refractive index detector Waters 2414 (RI) and a Wyatt DAWN EOS MALLS detector (**B**) The UPF were dissolved in ammonia, mixed with PMP, and neutralized with 200 µL of formic acid. The derivatization chromatomap was collected by UPLC/Q-TOF-MS. (**C**) Derivatization chromatomap of standard monosaccharides (Man, Rib, Rha, GluUA, GalUA, Glc, Gal, Xyl, Ara, Fuc).

**Figure 2 marinedrugs-17-00660-f002:**
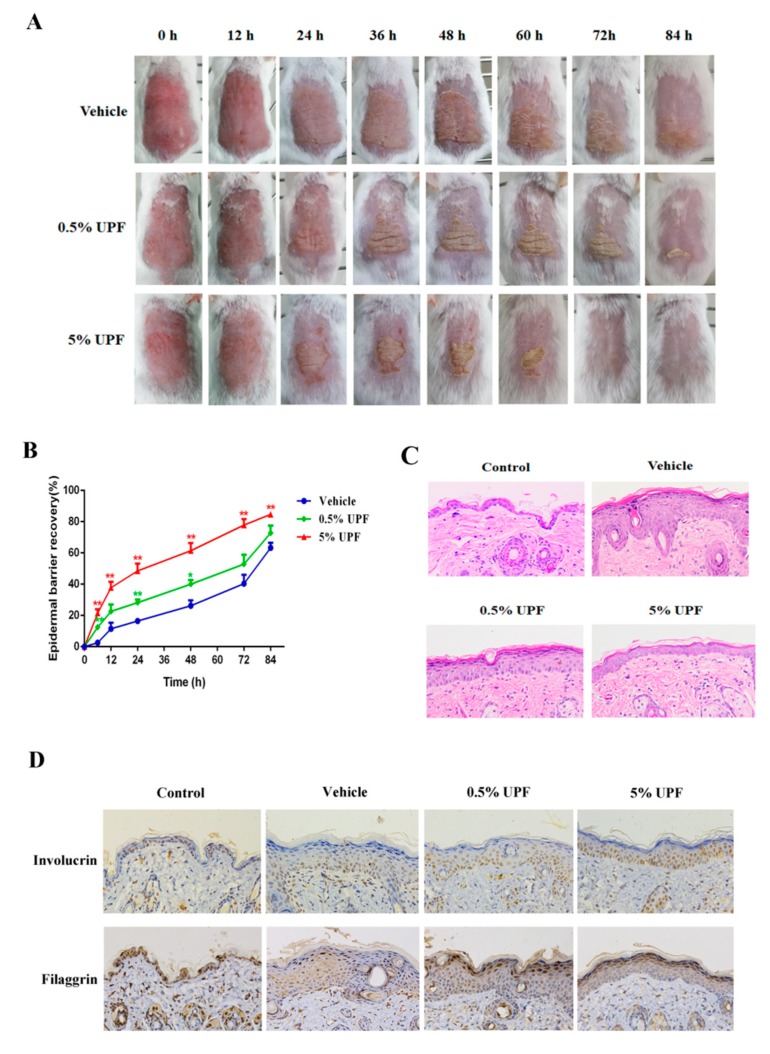
The effect of UPF on the recovery of epidermal barrier. Epidermal barrier disruption of ICR mice was induced by tape stripping on their shaved back skin until the TEWL reached 40 mg/cm^2^/hour. UPF hydrogel was administrated topically on the dorsal skin. (**A**) The photos were taken every 12 h after disruption. (**B**) The values of TEWL were measured at 0 h, 12 h, 24 h, 48 h, 72 h, 84 h. Data are presented as means ± SEM obtained from three independent experiments, * *p* < 0.05 and ** *p* < 0.01 versus the vehicle control. (**C**) The back skin were harvested at 84 h and the skin sections were prepared and stained with hematoxylin–eosin. (**D**) The back skin were harvested at 84 h and the skin sections were prepared and stained with immunohistochemistry.

**Figure 3 marinedrugs-17-00660-f003:**
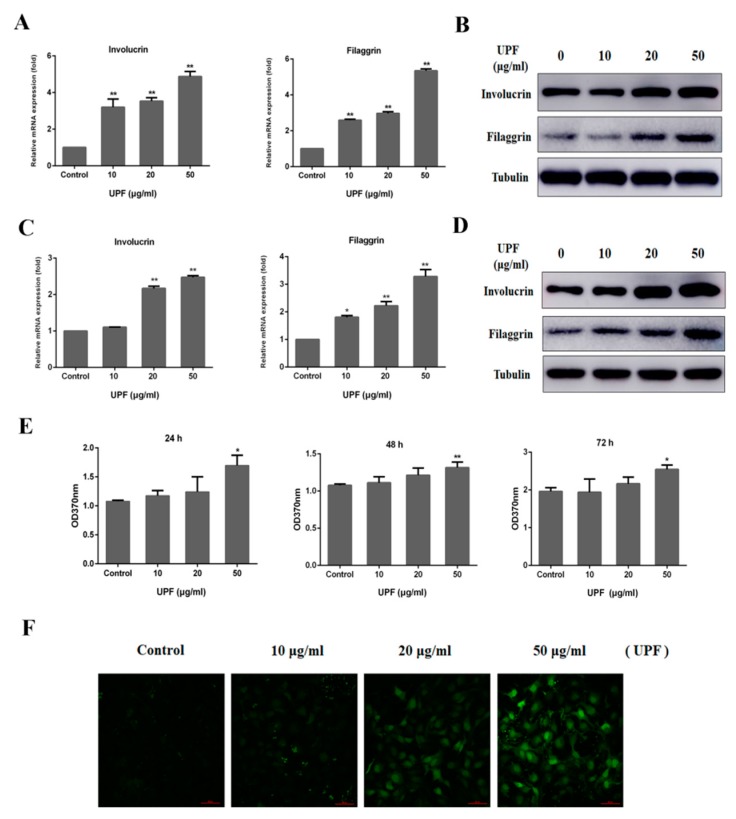
The effect of UPF on proliferation, differentiation and sustained Ca^2+^ concentration in HaCaT cells. HaCaT cells were treated with or without indicated concentrations (10, 20, 50 μg/mL of UPF. (**A**) The involucrin and filaggrin mRNA levels in HaCaT cells were assayed after 72 h treatment by qPCR. (**B**) The involucrin and filaggrin proteins levels in HaCaT cells were assayed after 72 h treatment by Western blot. (**C**) The involucrin and filaggrin mRNA levels in NHEK cells were assayed after 72 h treatment by qPCR. (**D**) The involucrin and filaggrin proteins levels in NHEK cells were assayed after 72 h treatment by Western blot. (**E**) The proliferation of HaCaT cells were assayed after 24 h, 48 h or 72 h treatment by BrdU assay. (**F**) HaCaT cells were treated with or without indicated concentrations (10, 20, 50 µg/mL) of UPF for 24 h, loaded with Fura-4 AM, and the fluorescence was recorded using a confocal microscope. Data are presented as means ± SEM obtained from three independent experiments, * *p* < 0.05 and ** *p* < 0.01 versus Control.

**Figure 4 marinedrugs-17-00660-f004:**
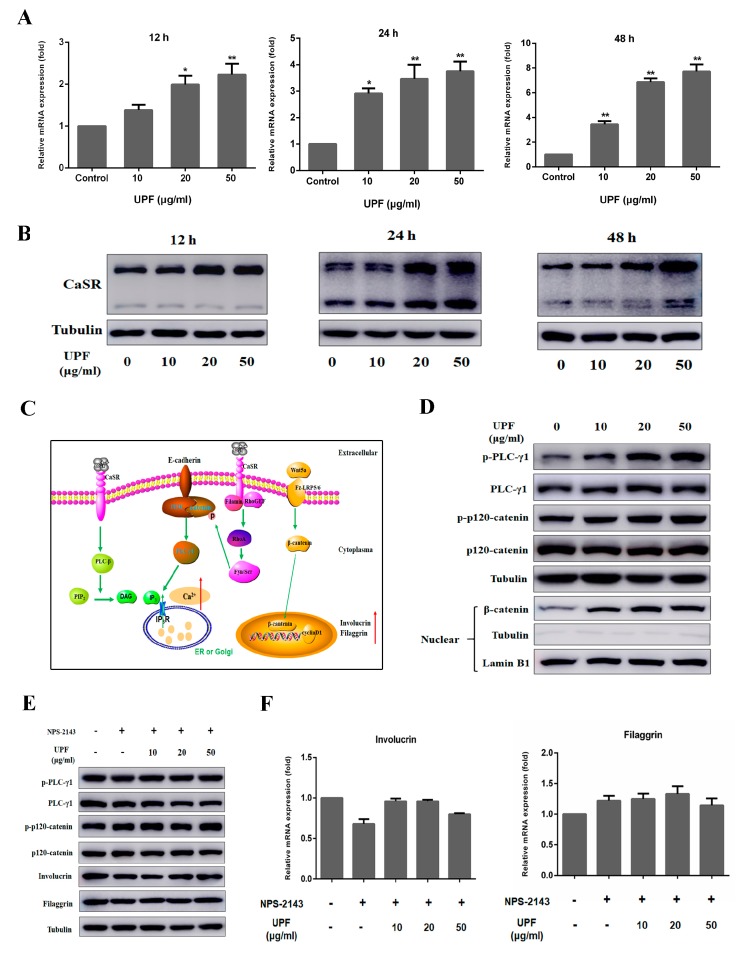
The effect of UPF on the expression of CaSR and CaSR mediated signaling pathway. (**A**) HaCaT cells were treated with or without indicated concentrations (10, 20, 50 µg/mL) of UPF, the CaSR mRNA levels in HaCaT cells were assayed at 12 h, 24 h, and 48 h by qPCR. (**B**) HaCaT cells were treated with or without indicated concentrations (10, 20, 50 µg/mL) of UPF, the CaSR protein levels in HaCaT cells were assayed at 12 h, 24 h, and 48 h by Western blot. (**C**) CaSR mediated signaling pathway. (**D**) HaCaT cells were treated with or without indicated concentrations (10, 20, 50 µg/mL) of UPF, the levels of PLC-γ1, p120-catenin, phospho-PLCγ1 and phospho- p120-catenin in cytoplasm and β-catenin in the nucleus were detected by Western blot. (**E**) HaCaT cells were exposed to indicate concentrations of UPF with or without NPS-2143 (CaSR inhibitor, 150 nM), the protein levels of PLC-γ1, p120-catenin, phospho-PLCγ1, phospho-p120-catenin, the involucrin and filaggrin were assayed by Western blot. (**F**) HaCaT cells were exposed to indicate concentrations of UPF with or without NPS-2143 (CaSR inhibitor) for 72 h, the involucrin and filaggrin mRNA levels were assayed by qPCR.

**Figure 5 marinedrugs-17-00660-f005:**
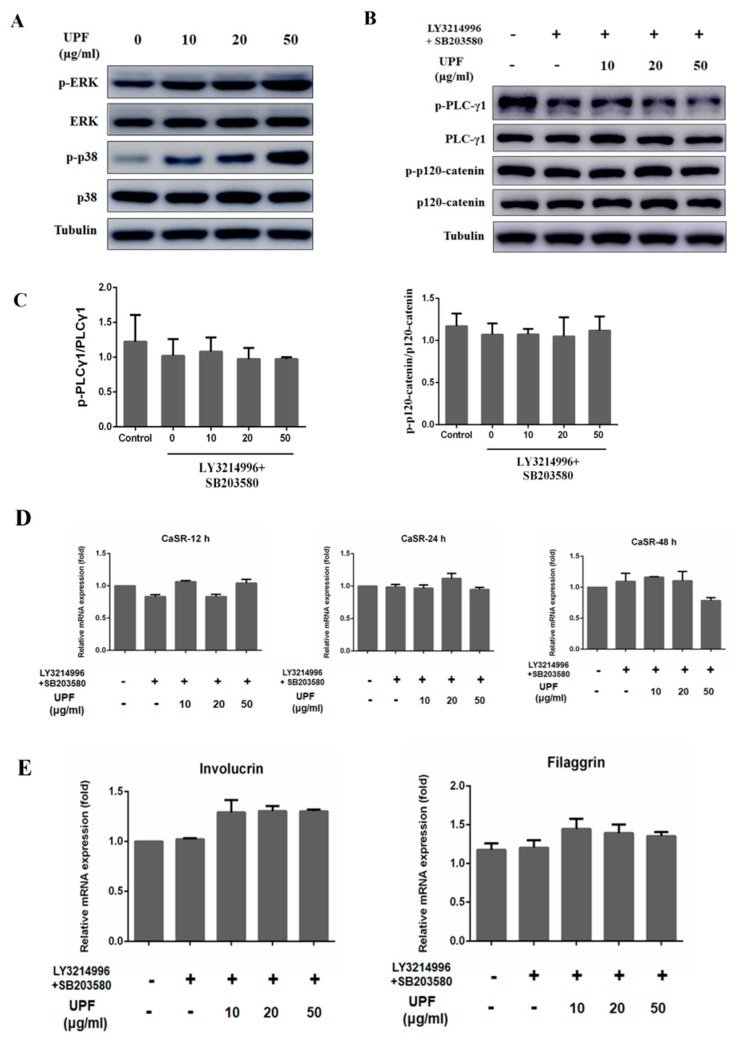
UPF increases the expression of CaSR via the ERK and p38 signaling pathway. (**A**) HaCaT cells were treated with or without indicated concentrations (10, 20, 50 μg/mL) of UPF, the levels of ERK, p38, phospho-ERK and phospho-p38 in cells were detected by Western blotting. (**B**) HaCaT cells were exposed to indicated concentrations of UPF with or without LY3214996 (ERK inhibitor, 100 nM) and SB203580 (p38 inhibitor, 20 µM) for 12 h, the levels of PLC-γ1, p120-catenin, phospho-PLCγ1 and phospho-p120-catenin in cells were detected by Western blotting. (**C**) The relative expression levels of phosphorylation of PLCγ1 and p120-catenin were detected by gray analysis. (**D**) HaCaT cells were exposed to indicated concentrations of UPF with or without LY3214996 and SB203580 for 72 h, the CaSR mRNA levels in HaCaT cells were assayed by qPCR. (**E**) HaCaT cells were exposed to indicated concentrations of UPF with or without LY3214996 and SB203580 for 72 h, the involucrin and filaggrin mRNA levels in HaCaT cells were assayed by qPCR.

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
