# Peer review of "Fucoidan from Undaria pinnatifida Ameliorates Epidermal Barrier Disruption via Keratinocyte Differentiation and CaSR Level Regulation"

_marinedrugs, 2019, doi:10.3390/md17120660_

Round 1
Reviewer 1 Report
i sugges to accept the ms
Reviewer 2 Report
This revised manuscript is addressed the all concerns and now acceptable for publication
Reviewer 3 Report
Authors addressed concerns of the reviewer by adding several new data to the revised manuscript. The reviewer suggests that their novel findings documented in the revised manuscript might well be important and deserve publication in the “Marine Drugs”.
This manuscript is a resubmission of an earlier submission. The following is a list of the peer review reports and author responses from that submission.
Round 1
Reviewer 1 Report
The ms investigates a very interesting topic. However, despite being widely used in the literature as a skin model, HaCaT cells do not show the same highly-proliferative, ‘basal’ cell, ‘wound-response’ phenotype that normal human epidermal keratinocytes (NHEK) demonstrate in serum-free, low Ca2+ medium. Therefore, the authors must either provide some data using NHEKs cells (for a select number of experiments to support HaCaT results), or at least make it explicitly clear (in the Abstract and Discussion) that the work was performed using this cell line.
Author Response
Response to Reviewer 1 Comments
Point 1: The ms investigates a very interesting topic. However, despite being widely used in the literature as a skin model, HaCaT cells do not show the same highly-proliferative, ‘basal’ cell, ‘wound-response’ phenotype that normal human epidermal keratinocytes (NHEK) demonstrate in serum-free, low Ca2+ medium. Therefore, the authors must either provide some data using NHEKs cells (for a select number of experiments to support HaCaT results), or at least make it explicitly clear (in the Abstract and Discussion) that the work was performed using this cell line.
Response 1: According to the review’s comment, we verify that UPF could promote the keratinocyte differentiation in NHEK cells, this result was showed in Figure 3C&D. We also supplied the reason for the selection of HaCaT cells for in vitro study of epidermal differentiation in the discussion.(Line, 238.) The spontaneously immortalized human HaCaT cell line is derived from normal adult skin and has been widely used in the study of keratinocytes and epidermal biology for its intact epidermal differentiation ability and similar biological characteristics to primary keratinocytes. What’s more, HaCaT is stable and easy to culture, so we select HaCaT as cell model in this research to study the effect of UPF on epidermal differentiation in vitro. This reason was provided to the discussion section.

Reviewer 2 Report
This paper describes the effects of polysaccharides, particularly fucoidan from Undaria Pinnatifida (UPF) on keratinocytes differentiation and function barrier repair. The authors also propose some mechanisms of action for these ingredient.
I appreciate the well documented introduction and experimentations showing that this molecule stimulates expression of filagrin and involucrin in HaCat cells and the proposition of some mechanisms of action such as augmentation of intracellular Ca under basal conditions. Treatment of keratinocytes with antagonists drugs on CaSR (calcium-sensitive receptor ), ERK, and P 38 suggest that UPF may modulate keratinocyte differentiation via these signaling pathway.
Indeed, to complete these study, I suggest others experiments in human ex vivo models of skin explants, because experiments were only conducted in mouse. For example, irradiated skin explants with UVs induced a great inhibition of differentiation markers, such as involucrin, loricrin, filagrin, claudin 1. It would be very interesting to test if UPF could restaure their expressions and consequently promote the recovery of barrier disruption in human.
Author Response
Response to Reviewer 2 Comments
This paper describes the effects of polysaccharides, particularly fucoidan from Undaria Pinnatifida (UPF) on keratinocytes differentiation and function barrier repair. The authors also propose some mechanisms of action for these ingredient.
I appreciate the well documented introduction and experimentations showing that this molecule stimulates expression of filagrin and involucrin in HaCat cells and the proposition of some mechanisms of action such as augmentation of intracellular Ca under basal conditions. Treatment of keratinocytes with antagonists drugs on CaSR (calcium-sensitive receptor ), ERK, and P 38 suggest that UPF may modulate keratinocyte differentiation via these signaling pathway.
Point 1: Indeed, to complete these study, I suggest others experiments in human ex vivo models of skin explants, because experiments were only conducted in mouse. For example, irradiated skin explants with UVs induced a great inhibition of differentiation markers, such as involucrin, loricrin, filagrin, claudin 1. It would be very interesting to test if UPF could restaure their expressions and consequently promote the recovery of barrier disruption in human.
Response 1: Thanks for reviewer’s instructive suggestions. In this study we aimed to clarify the repair effect of UPF on epidermal barrier damage due to low calcium environment in epidermis. We use tape-stripping method to destroy the epidermal barrier of mice, which leads to the increase of TEWL, causing the disappearance of calcium gradient in the epidermis, leading to a low-calcium environment. Actually, we have study the effect of UPF on epidermal barrier damage induced by UV radiation in another research , if we collect useful data, we will publish it.

Reviewer 3 Report
In the paper entitled "Fucoidan from Undaria pinnatifida ameliorates epidermal barrier disruption via keratinocyte differentiation and CaSR level regulation" present the data showing that Fucoidan from Undaria pinnatifida enhanced keratinocyte differentiation via CaSR, accelerating the repair of skin barrier.
There are weaknesses as detailed below.
In figure 2, authors assessed the effect of fucoidan on the changes of gross features, histologic features, and TEWL. Moreover, I suggest adding the expression of proteins about keratinocyte differentiation and skin barrier in mouse skin. Considering the involvement of the Wnt pathway in the proliferation and differentiation of epithelial cells inthe skin, I suggest to evaluate the effect of UPF on the wnt transcriptional activity. Line 215 in discussion section, the authors demonstrated that the UPF could promote both the proliferation and differentiation of keratinocytes under 1.8 mM calcium concentration. However, differentiation and proliferation do not go together.
Author Response
Response to Reviewer 3 Comments
In the paper entitled "Fucoidan from Undaria pinnatifida ameliorates epidermal barrier disruption via keratinocyte differentiation and CaSR level regulation" present the data showing that Fucoidan from Undaria pinnatifida enhanced keratinocyte differentiation via CaSR, accelerating the repair of skin barrier.
There are weaknesses as detailed below.
Point 1:In figure 2, authors assessed the effect of fucoidan on the changes of gross features, histologic features, and TEWL. Moreover, I suggest adding the expression of proteins about keratinocyte differentiation and skin barrier in mouse skin.
Response 1: Thanks for reviewer’s suggestion, we performed immunohistochemistry experiments in vivo to further investigate the UPF's ability to promote the recovery of barrier disruption. The result of immunohistochemistry showed that UPF could promote the protein expression of differentiation markers such as involucrin and filaggrin in epidermis, this result is consistent with the in vitro experiments. This result was supplied in the revised Figure 2D.
Point 2: Considering the involvement of the Wnt pathway in the proliferation and differentiation of epithelial cells in the skin, I suggest to evaluate the effect of UPF on the wnt transcriptional activity.
Response 2: Thanks for reviewer’s suggestion, we evaluated the effect of UPF on the wnt transcriptional activity by examining the protein expression of β-catenin in the nucleus by nuclear separation experiments that the HaCaT cell were treated with or without UPF. We found that UPF could promote the entry of β-catenin into the nucleus. The results are showed in the revised Figure 4D.
Point 3: Line 215 in discussion section, the authors demonstrated that the UPF could promote both the proliferation and differentiation of keratinocytes under 1.8 mM calcium concentration. However, differentiation and proliferation do not go together.
Response 3: It is necessary to clarify here that this promotion is not carried out in the same experiment. We performed qPCR and western blot to investigate the differentiation of keratinocytes under 1.8 mM calcium concentration at 72h, while the proliferation of keratinocytes was detected by BrdU assay at 24 h, 48 h or 72 h, as the manuscript says “the promotion of HaCaT cell proliferation by UPF in proportion to the dose of the latter at 24 h, but with no effect at 48 h and 72h at the tested concentrations”.(Line, 119. ) Please refer to the results in the revised figure 2A & B, figure 2E.

Reviewer 4 Report
In this manuscript, Chen et al. showed that UPF promoted TEWL recovery and attenuated epidermal hyperplasia after tape stripping in mouse skin. Authors also showed that UPF induced expression of differentiation markers and BrdU incorporation in HaCaT cells. Mechanistically, UPF induced expression of CaSR through activation of p38 MAPK and ERK, leading to enhanced [Ca2+]i elevation and differentiation. The manuscript contains several new findings. However, several key experiments are needed to support their conclusion. In particular, authors should confirm that in vitro results are also true under in vivo condition (i.e. mouse skin).
Major points
It is very important to show that UPF promotes barrier recovery by upregulating CaSR in mouse skin after barrier disruption. Authors should confirm that UPF treatment induces CaSR expression in vivo as well as in vitro HaCaT cell model. Although differentiation-inducing stimuli inhibit proliferation of normal keratinocytes, UPF induces both differentiation (Fig.3A) and proliferation (Fig.3B) in HaCaT cells. The reviewer is afraid that HaCaT cell is not an appropriate model for analyzing differentiation and proliferation of keratinocytes. Authors should perform experiments with primary or early passaged keratinocytes isolated from mouse or human epidermis. At minimum, the effects of UPF on mRNA and protein expression of differentiation markers and CaSR should be examined in primary or early passaged kerayinocytes. Authors should examined protein levels of key downstream targets of UPF, including differentiation markers (Fig.3A) and CaSR (Fig. 4A and D). How do p38 MAPK and ERK induce CaSR expression? Several control experiments should be performed. (1) In Fig. 4D, authors should confirm that solvent (vehicle) for NPS-2143 has no effect on expression of differentiation markers. mRNA expression of involucrin and filaggrin in cells treated with UPF and solvent (vehicle) for NPS-2143 should be measured along with other samples. In addition, the information on concentration of NPS-2143 should be provided. (2) In Fig. 5C, authors should confirm that solvent (vehicle) for LY3214996+SB203580 has no effect on expression of differentiation markers. mRNA expression of CaSR in cells treated with UPF and solvent (vehicle) for LY3214996+SB203580 should be included along with other samples. In addition, the information on concentration of LY3214996+SB203580 should be provided. In Fig.5B, authors mentioned that UPF-induced phosphorylation of p120 catenin was inhibited by treatment with LY3214996+ SB203580. However change in p-p120-catenin was not clear from the image of western blot. Quantification of western blot should be performed. In Fig.2B, are there no statistically significant effects of 5% UPF on barrier recovery comparing vehicle control?
Minor points
In abstract, CaSR is not an abbreviation for calcium-sensitive receptor but for calcium-sensitive receptor. In legend of Fig.2B, please correct TWEL to TEWL.Author Response
Response to Reviewer 4 Comments
Point 1: In this manuscript, Chen et al. showed that UPF promoted TEWL recovery and attenuated epidermal hyperplasia after tape stripping in mouse skin. Authors also showed that UPF induced expression of differentiation markers and BrdU incorporation in HaCaT cells. Mechanistically, UPF induced expression of CaSR through activation of p38 MAPK and ERK, leading to enhanced [Ca2+]i elevation and differentiation. The manuscript contains several new findings. However, several key experiments are needed to support their conclusion. In particular, authors should confirm that in vitro results are also true under in vivo condition (i.e. mouse skin).It is very important to show that UPF promotes barrier recovery by upregulating CaSR in mouse skin after barrier disruption. Authors should confirm that UPF treatment induces CaSR expression in vivo as well as in vitro HaCaT cell model.
Response 1: Thanks for reviewer’s thoughtful comments. We performed immunohistochemistry experiments in vivo to further investigate the UPF's ability to promote the recovery of barrier disruption. The result of immunohistochemistry showed that UPF could promote the protein expression of differentiation markers such as involucrin and filaggrin in epidermis, this result is consistent with the in vitro experiments. This result was supplied in the revised Figure 2D.
Point 2:Although differentiation-inducing stimuli inhibit proliferation of normal keratinocytes, UPF induces both differentiation (Fig.3A) and proliferation (Fig.3B) in HaCaT cells. The reviewer is afraid that HaCaT cell is not an appropriate model for analyzing differentiation and proliferation of keratinocytes. Authors should perform experiments with primary or early passaged keratinocytes isolated from mouse or human epidermis. At minimum, the effects of UPF on mRNA and protein expression of differentiation markers and CaSR should be examined in primary or early passaged kerayinocytes.
Response 2: We agree with the reviewer that the necessary experiments should be performed on primary or early passaged kerayinocytes. We verify that UPF could promote the keratinocyte differentiation in NHEK cells, this result was showed in Figure 3C&D. The spontaneously immortalized human HaCaT cell line is derived from normal adult skin and has been widely used in the study of keratinocytes and epidermal biology for its intact epidermal differentiation ability and similar biological characteristics to primary keratinocytes. What’s more, HaCaT is stable and easy to culture, so we select HaCaT as cell model in this research to study the effect of UPF on epidermal differentiation in vitro. This reason was provided to the discussion section.(Line, 238.)
Point 3: Authors should examined protein levels of key downstream targets of UPF, including differentiation markers (Fig.3A) and CaSR (Fig. 4A and D).
Response 3:We have examined protein expression levels of CaSR and differentiation markers including involucrin and filaggrin , those results were showed in figure 3B, figure 3D, 4B & E.
Point 4: How do p38 MAPK and ERK induce CaSR expression? Several control experiments should be performed.(1) In Fig. 4D, authors should confirm that solvent (vehicle) for NPS-2143 has no effect on expression of differentiation markers. mRNA expression of involucrin and filaggrin in cells treated with UPF and solvent (vehicle) for NPS-2143 should be measured along with other samples. In addition, the information on concentration of NPS-2143 should be provided. (2) In Fig. 5C, authors should confirm that solvent (vehicle) for LY3214996+SB203580 has no effect on expression of differentiation markers. mRNA expression of CaSR in cells treated with UPF and solvent (vehicle) for LY3214996+SB203580 should be included along with other samples. In addition, the information on concentration of LY3214996+SB203580 should be provided.
Response 4:
Thanks for reviewer’s suggestions, the solvent used in NPS-2143, LY3214996 and SB203580 is DMSO, but the content of DMSO in the culture system is not more than 0.1%, so we think this effect could be ignored.
In addition, the concentrations of these three inhibitors NPS-2143, LY3214996 and SB203580 in this study were 150 nM, 100 nM, 20 μM, respectively, which we have supplemented in the revised Figure 4E and Figure 5B legends, respectively.
According to the reviewer's suggestion, in Figure 4E we demonstrated the effect of UPF on CaSR mediated pathway and protein expression of involucrin and filaggrin once the CaSR was blocked. In addition, UPF could not elevate the mRNA level of the involucrin and filaggrin once the ERK and P38 signaling pathway was blocked. The new results are shown in Figure 5E.
Point 5: In Fig.5B, authors mentioned that UPF-induced phosphorylation of p120 catenin was inhibited by treatment with LY3214996+ SB203580. However change in p-p120-catenin was not clear from the image of western blot. Quantification of western blot should be performed.
Response 5: According to the reviewer’s comments, we analyzed the relative phosphorylation degree of PLCγ1 and P120-catenin by gray scanning. This result was presented in the revised Figure 5C.
Point 6: In Fig.2B, are there no statistically significant effects of 5% UPF on barrier recovery comparing vehicle control?
Response 6: Thanks for your careful work, we have compared the 0.5 % and 5 % UPF groups with vehicle group . The barrier recovery rate of the 0.5 % and 5 % groups were significantly accelerated compared with vehicle control before 72 h, only 5 % UPF treated mice showed significant improvement in repair rate at 72 h and 84 h. This result was presented in figure 2B.
Point 7: In abstract, CaSR is not an abbreviation for calcium-sensitive receptor but for calcium-sensitive receptor. In legend of Fig.2B, please correct TWEL to TEWL.
Response 7:Thanks for your careful work, we have corrected these mistakes and careful recheck.
